# Role of Effective Policy and Screening in Managing Pediatric Nutritional Insecurity as the Most Important Social Determinant of Health Influencing Health Outcomes

**DOI:** 10.3390/nu16010005

**Published:** 2023-12-19

**Authors:** Hema Verma, Arun Verma, Jeffery Bettag, Sree Kolli, Kento Kurashima, Chandrashekhara Manithody, Ajay Jain

**Affiliations:** 1SLU College for Public Health and Social Justice, Saint Louis University, Saint Louis, MO 63104, USA; 2Department of Pediatrics, Saint Louis University School of Medicine, Saint Louis, MO 63103, USA; arun.verma@health.slu.edu (A.V.); jeffery.bettag@health.slu.edu (J.B.); sree.kolli@slu.edu (S.K.);

**Keywords:** Social Determinants of Health, food insecurity, microbiota

## Abstract

Social Determinants of Health (SDOH) impact nearly half of health outcomes, surpassing the influence of human behavior, clinical care, and the physical environment. SDOH has five domains: Economic Stability, Education Access and Quality, Health Care Access and Quality, Neighborhood and Built Environment, and Social and Community Context. Any adversity arising out of these interlinked domains predominantly affects children due to their greater susceptibility, and the adverse outcomes may span generations. Unfavorable SDOH may cause food insecurity, malnutrition, unbalanced gut microbiome, acute and chronic illnesses, inadequate education, unemployment, and lower life expectancy. Systematic screening by health care workers and physicians utilizing currently available tools and questionnaires can identify children susceptible to adverse childhood experiences, but there is a deficiency with respect to streamlined approach and institutional support. Additionally, current ameliorating supplemental food programs fall short of pediatric nutritional requirements. We propose a nutrition-based Surveillance, Screening, Referral, and Reevaluation (SSRR) plan encompassing a holistic approach to SDOH with a core emphasis on food insecurity, coupled with standardizing outcome-based interventions. We also propose more inclusive use of Food Prescription Programs, tailored to individual children’s needs, with emphasis on education and access to healthy food.

## 1. Introduction

According to Healthy People Framework 2030, which was conceived by the US National Secretary’s Advisory Committee on National Health Objectives for the year 2030, Social Determinants of Health (SDOH) can be defined as “The conditions in the environments where people are born, live, learn, work, play, worship, and age that affects a wide range of health, functioning, and quality of life outcomes and risks.” [1] The SDOH has been categorized into five domains: Economic Stability, Education Access and Quality, Health Care Access and Quality, Neighborhood and Built Environment, and Social and Community Context [1]. Table 1 indicates the domains and empirically identifies the dimensions and subdimensions of SDOH. The influence of SDOH varies among people and is often interlinked. For instance, education and transportation access can shape employment opportunities, while one’s residence can impact access to a balanced diet. These factors have a variable distribution among different social strata that generate preventable and inequitable health disparities across diverse population segments. Further, the substantial impact of SDOH can persist across generations and foster health disparities stemming from factors like race, ethnicity, and socioeconomic status [2].

Social and structural elements exert a noteworthy influence on uneven health consequences. As indicated by an official evaluation by the Office of Assistant Secretary for Planning and Evaluation of the US Government, only 20% of health disparities at the county level stem from clinical care, whereas SDOH contributes to around 50% of health results [3]. SDOH alone can therefore substantially impact 47% of health outcomes. The remaining 53% are attributed to health behaviors (34%), clinical care (16%), and the physical environment (3%), respectively [3]. Another report by the Economic Research Service of the US Department of Agriculture noted that addressing food insecurity is crucial in improving population health, as it is a major social determinant of well-being [4]. Therefore, hospitals must consider behavioral, socioeconomic, and environmental factors when implementing a comprehensive health strategy. Although SDOH encompasses many factors, this paper primarily focuses on food insecurity and its interconnected facets. Table 1 lists the SDOH domains.

**Table 1 nutrients-16-00005-t001:** Empirically identified SDOH domain dimensions.

Domains	Dimensions	Subdimensions
**Economic Stability**	Economic indicators	Income, Assets, Assistance Programs, Indebtedness
	Employment	Unemployed, Migrant/seasonal, Day laborers, Disability status, Retirement status, Student, Job assistance
	Material hardship	Food insecurity, Utilities, Transportation Medication affordability, Access to technology, Childcare, Clothing, Legal services
**Education Access and Quality**	Education	Educational attainment, Basic literacy, Health literacy, Numeracy
	Language	Primary language, English proficiency, Interpreter/translator needed, other language proficiency
**Health Care Access and Quality**	Functional status	ADLs, IADLs, Frailty
	Health behaviors	Alcohol, Drug use, Tobacco, Secondhand smoking, Physical activity, Sexual activity, Safety Diet
	Healthcare access	Insurance status, Healthcare affordability, Source of usual care
	Mental health	Depression, Anxiety, PTSD, ADD/ADHD, Suicide/Self-harm, Stress, Sleep
**Neighborhood and Built Environment**	Demographics	Gender/sexual orientation, Place of birth, Race/ethnicity, Refugee status, Justice Involvement
	Housing	Homelessness, Housing safety, Housing quality, Housing insecurity
**Social and Community Context**	Culture	Religion/spiritual beliefs, Family culture
	Family	Marital status, Dependents, Living arrangements
	Social support	Community activities, Safe environment, public spaces, Racism, Discrimination, Trust School culture, social isolation
	Trauma/violence	IPV, Trauma, Physical abuse, Sexual abuse, Mental abuse
	Veteran status	Military trauma history, Combat veteran, Active Military

After Byhoff E et al. [5]. ADD/ADHD: Attention deficit disorder/hyperactivity disorder; ADL: activity of daily living; IADL: independent activities of daily living; IPV: interpersonal violence; PTSD: Post-traumatic stress disorder; SDOH: Social and Behavioral Determinant of Health.

## 2. Food Insecurity and Associated Factors

### 2.1. Food Insecurity

According to the US Department of Agriculture, “Food insecurity is the limited or uncertain availability of nutritionally adequate and safe foods or limited or uncertain ability to acquire acceptable foods in socially acceptable ways” [6]. Food insecurity is influenced by factors beyond individual household attributes like income, employment status, and household composition. At the state level, average wage levels, housing costs, unemployment rates, and the impact of state-level regulations on access to unemployment insurance, the income tax credit, and nutritional support initiatives are some of the contributing factors [7]. Food insecurity in households with children can lead to frequent illnesses, longer recovery time, increased hospitalizations, poor school performance, and emotional/behavioral issues [8]. According to data from 2021, the average American household spent $62.50 per person each week on food. Compared to the Thrifty Food Plan (TFP), which adjusts for inflation and considers people’s specific food needs based on age and gender, the median household spending was 1.15 times the estimate. A ratio of household food spending over 1.0 means they spent more than the TFP, while a ratio below 1.0 means they spent less. Therefore, in 2021, the typical household spent 15% more on food than the cost of the TFP for their specific household [7]. This suggests a significant rise in the load on a low-income household for providing food. Therefore, on already financially strained families struggling with food insecurity, increased adverse health events associated with food insecurity further exacerbate financial hardship.

Income is a crucial determinant of health. It affects living conditions, well-being, and dietary practices. Low-income households may struggle to afford nutrient-rich foods, leading to the consumption of energy-dense and low-nutrient options (fast/junk food). Uncertain incomes can lead to short-term solutions and cost-reduction strategies that overlook self-care and long-term health considerations. Tough dilemmas about food may arise, such as choosing between medication and food or feeding oneself or one’s children [9]. Housing quality also affects health and well-being, with safety, crowding, and security playing a role. Stable and secure housing can improve diet and decrease food insecurity.

In contrast, food insecurity and temporary housing can negatively affect health, leading to a higher risk of depression and physical illness. In this scenario, public housing subsidies can help low-income families avoid undernutrition [9]. The food environment combines physical, economic, policy, and socio-cultural factors that impact people’s food and drink access, choices, and nutritional status. Accessing retail stores and food aid resources is a challenge for low-income individuals. Buying food can be difficult in low-income areas due to unaffordability and poor quality. Food aid outlets have limited hours and require referrals, fees, or subscriptions, making accessing them demanding. Additionally, availability depends on local resources, which creates uncertainty. Some rural and coastal areas need more resources to help, exacerbating the issue [9].

Between 2019 and 2020, 10.8% of children aged 0–17 were part of households facing food insecurity within the last 30 days. In 2021, 6.2% of households with children, comprising one or more children, experienced food insecurity. Moreover, the percentage of households with food-insecure children was notably higher for households led by females (12.1%), households with reference persons identifying as black and non-Hispanic (12.0%) or Hispanic (9.7%), households with incomes below 185% of the poverty line (14.5%), and households situated in principal cities (7.7%) [7]. Concerning urban living, children residing in large central metropolitan areas (13.2%) were more likely to live in homes affected by food insecurity than their counterparts in medium and small metropolitan areas (10.5%). Lastly, children aged 0–17 who lived in families with three or more children (13.0%) exhibited a higher probability of experiencing food insecurity in comparison to children in families with fewer than three children (9.4%) [10]. 

The COVID-19 pandemic, most notably in 2020, increased food insecurity and poverty among vulnerable communities, especially those in informal labor. Racism, discrimination, unstable conditions, chronic illnesses, and previous infections worsened the situation and outcomes [11]. As the globally used Integrated Food Security Phase Classification (IPC) system indicated, globally, more than 820 million people faced food insecurity during the pandemic’s peak. Among them, 135 million people were at crisis and emergency levels. As per the World Food Program estimates, approximately 130 million more were expected to join them by the end of 2020. Since February 2020, up to an estimated 45 million people may have been compelled to deal with acute food insecurity due to deteriorating employment conditions and associated factors [12].

### 2.2. Food Landscaping

*Food deserts* are areas where people have limited access to healthy food options, often due to financial or transportation barriers. *Food swamps* have adequate healthy options, but also an overabundance of unhealthy choices. Some low-income individuals may experience *food mirages* (unaffordability amid overabundance), making it difficult to access healthy, affordable food in their neighborhood [13]. One of the studies analyzed geocoded data from the California Healthy Kids Survey between 2002 and 2005, studying over 500,000 youths to determine the correlation between adolescent obesity and the proximity of fast-food restaurants to schools. Results showed that students whose schools were located within half a mile of fast-food restaurants consumed fewer servings of fruits and vegetables and more servings of soda, and were more likely to be overweight or obese than those whose schools were not near fast-food restaurants [14].

A similar study in Belo Horizonte City, Brazil, looked at food establishments and classified neighborhoods as food deserts or swamps. Food deserts had worse services, lower income, and fewer literate people compared to food swamps [15].

## 3. Effect of Food Insecurity and Associated Factors on Children

### 3.1. Development and Transition

SDOH wield an impact on the well-being of individuals spanning various age groups. However, the significance of SDOH is notably pronounced for children and young individuals. This is because the foundational elements for lifelong health and well-being, encompassing physical, social, and emotional capacities, are established during the early stages of life [16].

The probability of Non-Communicable Diseases (NCDs), such as heart disease, cancer, chronic respiratory disease, and diabetes, can be represented through a model based on the ratio of metabolic load to metabolic capacity called ‘the capacity-load model.’ When capacity is kept constant, an increase in load is anticipated to escalate NCD risk in a dose-response manner. Similarly, with the load held steady, a decrease in capacity is projected to amplify risk in a dose-response manner. The highest risk of NCDs is foreseen in individuals with reduced capacity and heightened load. This suggests that insufficient postnatal growth continues to curtail the development of metabolic capacity during early infancy, a phase marked by hyperplasic growth. In high-income countries, those from lower socioeconomic backgrounds have higher rates of chronic diseases, leading to premature mortality and more years of ill health. Disadvantaged groups often experience suboptimal nutrition and growth in early years, which can persist into adulthood, making them vulnerable to NCD risks. The capacity-load model helps illustrate the complex interplay of social determinants contributing to health inequalities [17].

An analysis of how Adverse Childhood Experiences (ACEs) impact pediatric health outcomes reveals evidence of hindered physical growth, cognitive advancement, elevated probabilities of childhood obesity, asthma, infections, non-febrile illnesses, disrupted sleep, delayed onset of menarche, and vague physical complaints [18]. Food-insecure families have kids with poorer health, including anemia, asthma, stomach issues, headaches, and colds. Nutritional deficiencies, obesity, and hospitalization risks are high among these children. Additionally, they struggle with cognitive, emotional, and behavioral problems [19].

### 3.2. Impact of Family and Social Conditions

Bronfenbrenner’s ecological systems theory suggests that individuals develop complex connections with their environment. He identified four systems in his model: microsystem, mesosystem, ecosystem, and macrosystem. The framework helps to comprehend and interpret diverse aspects of a child’s environment. It prompts us to consider factors that impact a child’s life beyond the immediate family sphere [18].

A twin study was conducted by the Centre for Epidemiology and Biostatistics, Melbourne School of Population and Global Health, The University of Melbourne, investigating the association between socioeconomic status and psychological distress. In this study, they utilized a twin design to account for genetics and environment in examining the association between socioeconomic status (SES) and psychological distress. To assess causality, they performed a between-pair regression analysis. SES was measured using various indices, while psychological distress was measured with the Kessler 6 Psychological Distress Scale. Findings demonstrated that low SES is linked to poor mental health, highlighting the importance of addressing social determinants alongside individual interventions [20].

The human microbiome, which is being studied as a factor influencing health, is shaped by social and geographical circumstances. Children living in resource-constrained environments are underrepresented in microbiome research, highlighting the impact of social disparities on health from an early age [16]. The effect of environmental and socioeconomic factors on the gut microbiome of healthy school-age children was studied. The study analyzed the gut microbiome of 176 Israeli Arab children, aged 6–9, from 3 villages with different socioeconomic statuses. The results indicated that a child’s gut microbiome is affected by their home’s socioeconomic status and level of crowding, leading to differences in bacterial diversity and metabolism [21].

The CDC (Centers for Disease Control) has funded and collaborated with Kaiser Permanente since 1994 on a study on ACEs labeled The ACE (Adverse Childhood Experiences) Study. The ACE Pyramid propounded by this study demonstrates the impact of ACEs in causing social, emotional, and cognitive impairment, promoting high-risk behaviors, and leading to disease, disability, social problems, and even early death. By understanding the childhood origins of myriad health and social issues across the lifespan, this study shows that prevention of ACEs such as abuse and neglect can enormously impact our society’s physical and psychosocial health. ACEs have been noted to have a steady dose-response relationship, and those with more ACEs or a higher ACE score (Range 0—10) have a higher likelihood of adverse health outcomes in childhood and adulthood [22].

### 3.3. Impact of Sociobiome

Socioeconomic factors may more significantly influence microbiome composition in children than adults, as a person’s microbiota is mainly established within the first three years of life, that is from birth until the incorporation of the adult diet. Aside from potential therapeutic implications, modifying the microbiota during childhood could be crucial in protecting against infections, allergies, asthma, and childhood cancers [16].

Sociobiome can be defined as the microbiota composition of a geographic region or neighborhood as a result of exposure to similar socioeconomic factors, which determine an environment with analogous characteristics that shape the individual microbiota into remarkable resemblance. Addressing the sociobiome can improve targeted health policies for regions with varying realities and issues, instead of broad/blind interventions [23]. Although comprehensive investigations have not yet been conducted, family members may influence an infant’s gut microbiota, with siblings potentially playing a role. A study in the Netherlands found infants with siblings had more Bifidobacterium when one month old. Infants without older siblings had lower levels of facultative anaerobes and other anaerobes but higher levels of Clostridium and Escherichia coli. However, the “sibling effect” remains a topic of discussion, given its complex and challenging-to-quantify nature. Additionally, geographical location could indirectly impact early-life microbiota due to its influence on dietary habits and lifestyle [24]. 

The colon is one of the planet’s most densely populated microbial environments, containing over 100 trillion microorganisms [25]. In the gut, Bacteroidetes and Firmicutes are the dominant species in adults, while other phyla like Actinobacteria, Proteobacteria, and Verrucomicrobia are present in smaller numbers. Gut bacteria have several functions, including digestion, vitamin production, immune response regulation via metabolites, lipid metabolism, and protection from pathogens [26].

Pregnant women with complications have less diverse gut microbiota, negatively affecting both mother and fetus. Factors such as diet, antibiotics, infections, stress, immune status, age, and genetic makeup play a role. The period around childbirth and early infancy is vital for establishing gut microbiota in offspring. Factors such as mode of delivery, gestational age, feeding method, maternal diet, environment, and host genetics shape infant gut microbiota composition. Neonatal gut microbes form through the complex interplay between environmental factors and host variables. Cesarean sections can disrupt the natural development of an infant’s microbiota, contributing to reduced diversity. Premature births, antibiotics, and parenteral feeding can also harm an infant’s microbiota. Breastfeeding has a dual effect on the intestines of breast-fed infants, enhancing the relative abundance of Bifidobacterium and supplying newborns with maternal microbes, vital nutrients, and antibacterial agents. IgA in breast milk helps regulate the infant’s immune system [24].

The introduction of solid foods alters infant gut microbiota richness and diversity. Fermentation of dietary fiber by gut microbiota produces short-chain fatty acids (SCFAs), affecting host immunity and metabolism. Infants exhibit lower microbial diversity compared to stable adult gut microbiota. Human milk oligosaccharides (HMOs) act as natural prebiotics, relying on Bifidobacterium for absorption. A negative correlation exists between HMO levels in infant feces and Bifidobacterium abundance. HMOs lack direct nutritional value but shape infants’ gut microbiota for long-term health benefits [24].

The gut microbiome’s connection with various diseases lacks a universal marker for health. Genetics, exposome (“the cumulative measure of environmental influences and associated biologic responses throughout the life span, including exogenous exposures and endogenous processes”—a term coined by Dr. Christopher Wild, rearticulated by Dr. Gary W Miller), lifestyle, and diet affect microbiome in both well-being and illness. A Dutch study examined 8208 individuals from a multi-generational cohort in the Netherlands. They assessed bacterial composition, function, antibiotic resistance, and virulence factors in relation to 241 host and environmental factors, encompassing physical and mental health, medication use, diet, socioeconomic conditions, childhood, and current exposome. Their findings revealed that the environment and cohabitation predominantly shape the microbiome, and numerous factors from early life and the present are substantially associated with microbiome composition and function [27]. Therefore, it is essential to include children in clinical trials to evaluate dietary modifications and their effects on health and diseases [17].

## 4. SDOH Effect on the Nutrition of Children

### 4.1. Obesity and Malnutrition

Analysis of the National Survey of Children’s Health (NSCH) 2016–2017 dataset found that 31% of children surveyed were overweight or obese. Non-Hispanic Black and Hispanic children were more likely to be obese. Younger children, those with a single parent, and those in amenity-poor neighborhoods were more likely to be overweight. Parental college education, health insurance coverage, female gender, and speaking a language other than Spanish at home were protective factors against overweight/obesity [28]. This data showcases the disparity in susceptibility to these conditions based on factors like race, family structure, neighborhood amenities, and socioeconomic indicators.

According to research published in 2014, all age groups consume less dietary fiber (DF) than the recommended amount. On average, children and adolescents consume less than 14 g of DF daily, while adults consume around 17 g. Additionally, adults with lower family incomes and those living below the poverty line also consume less DF. As a solution, it has been suggested that federal and local government policies should encourage the consumption of all vegetables, including the white potato, as an essential source of DF [29]. These insights emphasize the need for a comprehensive approach that considers socioeconomic disparities, dietary habits, and policy interventions to address the complex challenge of childhood obesity and inadequate dietary fiber intake.

### 4.2. Gut Microbiota and Nutrition

Studies show that malnourished children in low-income areas are more likely to have pathogenic gut microbiota. While there is agreement on the link between nutrition and gut health, the effectiveness of modifying the human microbiome for therapeutic purposes is still debated. Fecal microbiota transplantation has proven helpful in specific cases, but using probiotics, prebiotics, and dietary interventions for treating infections or inflammatory diseases has produced inconsistent results [16].

The gut microbiota can enhance energy extraction from indigestible carbohydrates via novel receptor and enteric hormone mechanisms. Gut microbiota may influence lipid metabolism and adipogenesis, provoking the pathogenesis of obesity and metabolic disorders. These effects on humans require further study. Additionally, the gut microbiota and probiotics such as Lactobacillus and Bifidobacterium species generate B-group vitamins. Vitamin D and the gut microbiota act in sync to maintain calcium balance, potentially by augmented activity of the VDR and protein kinase signaling pathways. However, the role of the microbiota in vitamin K status requires further investigation. Systems biology approaches, including metagenomic and metabolomic techniques, can deepen our understanding of the gut microbiome’s interplay with the host genome in maintaining health equilibrium and nutritional well-being [30].

Malnutrition causes deficiencies in innate and adaptive immune systems, increasing vulnerability to diarrheal diseases. Repeated infections also compromise nutrient levels and intestinal mucosal barrier function, leading to a cycle of worsening health. A complicating factor is that vaccines intended to shield children against specific pathogens exhibit limited effectiveness in regions with pervasive malnutrition. A testable notion suggests that the gut microbiota might contribute to disease susceptibility and progression by affecting nutrient processing, absorption, and immune activity. Diet shapes the structure and function of the gut microbial community, while the microbiota adapts in ways that enhance nutrient processing. The microbiota’s ability to process a given diet influences its nutritional and energetic value. 

Furthermore, the microbiota and immune systems evolve in tandem: malnutrition impacts both the innate and adaptive immune systems and the microbiota. The microbiota is a barrier against enteropathogens, but malnutrition and immune system disturbances can disrupt this function. Additionally, microbiota affects nutrient processing and distribution, which can influence the expression of host genes responsible for nutrient transport and metabolism [31]. This data highlights the connection between SDOH, gut microbiota, and nutritional well-being (Figure 1). It underscores how SDOH factors can negatively impact children’s development. Figure 2 illustrates this intricate correlation and highlights the role of healthcare providers, including pediatric practitioners, nutritionists, and social workers at the organizational level in alleviating its effects.

## 5. Healthcare Workers’ Role in Screening, Liaison, Evaluation, and Management

Screening for unaddressed social and financial needs aligns with the American Academy of Pediatrics (AAP) directives. Child health professionals have a unique opportunity to address the social factors influencing children’s well-being and to prevent childhood adversity. However, they need the necessary tools and resources to do so effectively [19]. Child health professionals can address ACEs and harmful stress by understanding the prevalence of ACEs in their communities. Systematic screening can identify children who require additional resources such as food assistance, social support services, and government-backed nutrition programs [19].

Conversely, positive influences in childhood (PICs) can actively promote children’s physical and mental well-being as they mature into adulthood [19,32]. Studies examining their effectiveness have shown that interventions involving screening and referral for SDOH can enhance physicians’ awareness of the prevalence of these determinants. However, a lack of training, guidance, and support for healthcare providers hampers the early detection of ACEs and the connection of local healthcare services with food assistance programs [22,33]. Recently, the Centers for Medicare and Medicaid have started offering incentives to encourage innovation in the screening and addressing SDOH, further promoting their adoption in medical practices [5]. The CDC website provides a set of ACE questionnaires at no cost, empowering caregivers to screen for ACEs in both clinical and non-clinical settings, thus providing an opportunity for early intervention through counseling and support [31].

Surveillance in this context is “described as a flexible, ongoing process carried out by knowledgeable professionals while providing healthcare services.” In contrast, screening involves the use of standardized tools. In the context of surveillance and screening for social determinants, there are four essential components:Healthcare providers should routinely ask parents general questions to uncover their concerns and needs.Identifying any existing risk factors or protective factors is crucial.Specific social issues should be systematically screened during regular check-ups.Patients and families with identified needs should be referred to professionals in other fields and community organizations that can provide assistance and resources, such as the Medicaid office or legal advocacy groups.

A team-based approach can streamline the screening process. Many of these screenings can be conducted and evaluated by non-physicians and integrated into electronic health records. Patients and caregivers should be informed that the practice screens all patients universally to avoid any concerns about stigma [19].

Physicians recognize that social determinants such as income, education, and social status can impact their patients’ health. However, many are uncertain about how to address these factors. To act on SDOH at the patient, practice, and community levels, physicians and other healthcare professionals can:At the patient level, sensitively inquire about social challenges, refer patients, and assist them in accessing benefits and support services.At the practice level, improve access and the quality of care for hard-to-reach patient groups and incorporate patient social support navigators into the primary care team.At the community level, collaborate with community organizations, public health entities, and local leaders to create a healthier environment. Physicians can also advocate for social change by leveraging their clinical experience and research evidence, participating in community needs assessment and health planning, and engaging in community empowerment efforts and changing social norms [34].

## 6. Models and Tools Related to SDOH

The process of integrating advancements from clinical research into standard practice can span up to 17 years. The screening and referral approach for addressing SDOH is still in its early phases, lacking empirical evidence regarding optimal methodologies. Although healthcare providers acknowledge the significance of SDOH screening, challenges like time constraints and logistical issues impede the effective implementation of interventions. Therefore, it is essential to establish appropriate screening and referral processes within clinical teams to facilitate regular SDOH screening and referral [5]. SingHealth Community Hospitals in Singapore uses social prescribing to address the aging population’s complex health and social needs. The team took an iterative approach; the implementation team consistently reviewed and adjusted practices, work procedures, and outcome assessment tools based on data and input from stakeholders to tackle implementation challenges. Ongoing program evaluation will establish evidence-based best practices [35].

In 2022, the Centers for Medicare & Medicaid Services (CMS) introduced new quality measures to promote health equity, in collaboration with entities like the National Committee for Quality Assurance (NCQA) and The Joint Commission. These measures primarily focus on screening patients for health-related social needs (HRSN). However, a more sensitive approach is needed, focusing on marginalized patients and communities. Health systems should involve patients, staff, and community members in designing social care programs and advocate for health-promoting safety net initiatives. Policymakers and health systems should invest in addressing HRSN comprehensively to benefit patients and communities [36].

The American Academy of Pediatrics advises that SDOH should be assessed during every visit from infancy to age 21. Nevertheless, no universal screening tool can be applied due to differences in family requirements and resources. Customized screening by pediatric professionals is a potential solution, and the Eric B. Chandler Health Center in New Jersey designed a paper handout containing “yes-no” questions to assess the social needs of pediatric patients. This tool underwent continuous improvement through PDSA (plan-do-study-act) cycles. Positive screens were then directed to community health workers who offered resources and counseling [37].

The patient-centered medical home (PCMH) is a way to improve population health by systematically addressing SDOH. Pediatrics has been at the forefront of developing the medical home and continues to improve clinical practice to address social determinants, especially for children. PCMHs within the accountable care organization model can provide innovative services, including new financial models that incentivize adaptations for adult populations. In urban pediatric clinics, on-site community-based resources like WIC (Special Supplemental Nutrition Program for Women, Infants, and Children) are common, leading to benefits like addressing transportation challenges, streamlining community services, enhancing patient satisfaction, and improving access to suitable social services. The medical-legal partnership model integrates legal services within pediatric and specific adult healthcare settings to address the everyday legal needs of families that impact health, such as housing conditions and food security. Leveraging visual media and smartphone monitoring can enhance patient wellness behavior. Home visiting programs can also play a crucial role in helping physicians address their patients’ social determinants beyond the confines of the PCMH. These programs focus on promoting child development, parenting skills, and aiding parents with specific needs like school enrollment, employment, and accessing social safety net programs. Home visits provide healthcare professionals with valuable insights into the child’s living conditions [38]. Table 2 indicates some of the significant SDOH screening tools and models that can be utilized.

Health Leads (formerly Project HEALTH), a nonprofit organization founded in 1996 at the Boston Medical Center, places undergraduate students in urban clinics to assist underprivileged families with their social needs. Their innovative approach involves pre-visit screening surveys, provider referrals, and student connections to community-based resources, effectively reducing unmet social needs for low-income families. The multidisciplinary team-based approach acts as a bridge between medical homes and community resources. Health Leads’ success speaks volumes about the efficacy of their model [40].

A comprehensive family psychosocial history should be an integral part of well-child visits. The mnemonic “IHELLP” offers a strategy to assist providers in addressing family-related issues, including income, housing, education, legal status/immigration, literacy, and personal safety. The patient history should be updated regularly by asking, “Have there been any changes in your or your family’s needs since our last visit?” [41].

After screening and evaluating patients, pediatric patients who are malnourished should be referred to resources that meet their nutritional needs. Creating healthy food environments in communities is a complex task that requires collaboration across different sectors to improve the community’s health. The most practical method involves making healthy and affordable food options available while reducing the prevalence of less nutritious foods. To achieve this, public health organizations, local governments, businesses, food suppliers, and distributors must work together [13]. Healthcare institutions are implementing initiatives to provide access to fruits and vegetables through different program models, including vouchers, garden-based programs, and meal delivery. The most common model is fruit and vegetable vouchers, constituting 63% of these initiatives. Vouchers offer a designated value for fresh produce at specific locations [33]. Food Prescription Programs (FPPs) offer discounted or free vouchers for fruits and vegetables to those with food insecurity and chronic conditions. However, there is limited evidence on program outcomes and challenges with sustainability and inclusivity. To be effective, programs must balance access and education, and connect with clinical settings for long-term access. Standardizing expertise and clinical outcomes are crucial for optimizing food Rx programs for chronic conditions [33].

A recent study examined individuals who participated in a Fresh Food Prescription delivery program to assist those experiencing food insecurity during the COVID-19 pandemic. The study discovered that participants believed the program positively impacted their health and well-being. They also suggested employing culturally specific strategies for improvement. This study represents the first theory-based investigation in the United States [42].

## 7. Intervention Suggestions

All the models and strategies mentioned above prioritize the screening and guidance of families with pediatric patients based on their specific SDOH disparities. However, these plans lack the capability to continuously screen, refer, and reevaluate pediatric patients according to their nutritional needs. In light of both the nutritional necessities of pediatric patients and the organizations’ and healthcare providers’ roles, we propose a nutrition-based Surveillance, Screening, Referral, and Reevaluation (SSRR) plan that employs an iterative and agile approach. This plan integrates volunteer participation to alleviate the workload of pediatric healthcare practitioners. It is a modified version of the Healthy Leads model.

Patients visiting primary care providers must complete a pre-visit screening form that assesses SDOH-related disparities. The pediatrician then addresses the identified needs based on the patient’s nutritional requirements. Subsequently, the patient’s family is referred to the Healthy Needs staff. These staff members are trained volunteers who collaborate with social workers, nutritionists, and dieticians under the guidance of the pediatric practitioner.

The Healthy Needs staff conducts a comprehensive assessment using SDOH tools to pinpoint areas of deficiency. After discussions with the pediatric practitioner, nutritionist, and social worker, the pediatric patient’s family receives a personalized food prescription tailored to the patient’s nutritional needs. The volunteers guide patients to resources like food pharmacies. They also maintain follow-up communication with the families to track progress and update the physician.

The physician then reevaluates and reassesses the patient’s progress, adjusting the plan to accommodate changes. Figure 3 explains a comprehensive flowchart of the SSRR plan model.

## 8. Conclusions

It is vital to recognize and address SDOH to provide the best possible care for children. These factors include environmental conditions such as food insecurity, limited access to healthy food choices, adverse childhood experiences, and the impact of the gut microbiome on overall health and well-being. The complex interrelation between the SDOH and the gut microbiota significantly intensifies the nutritional impact during the early developmental stages. Unfortunately, the COVID-19 pandemic has only amplified these disparities, particularly in vulnerable communities. We need a comprehensive approach beyond traditional medical interventions to make a difference in pediatric healthcare. Healthcare providers must have the necessary tools to identify and address SDOH by screening for adverse childhood experiences, harmful stress, and other social issues.

Additionally, we must work in partnership with community organizations and local leaders to improve access to resources that meet the nutritional needs of pediatric patients. We have seen the success of social prescribing, patient-centered medical homes, and programs like Health Leads and Fresh Food Prescription in addressing SDOH in pediatric care. By implementing a Surveillance, Screening, Referral, and Reevaluation (SSRR) plan that leverages the Healthy Leads model, we can create personalized food prescriptions for pediatric patients according to their nutritional requirements. This approach recognizes the interconnectedness of social and environmental factors with health outcomes and emphasizes the importance of a holistic and proactive approach to pediatric healthcare.

It is crucial that we take steps to acknowledge and mitigate the impact of SDOH on pediatric nutrition and health. Working with healthcare providers, community organizations, and local leaders can create a more equitable and healthier future for our most vulnerable and youngest populations. Integrating these strategies into clinical practice and healthcare institutions represents a significant step in narrowing the gap in meeting SDOH-related nutritional needs, leading to improvements in health and the quality of life for children in our communities. It is important that we continue to sustain research and the implementation of these approaches as we strive toward a society that prioritizes equity and health.

## Figures and Tables

**Figure 1 nutrients-16-00005-f001:**
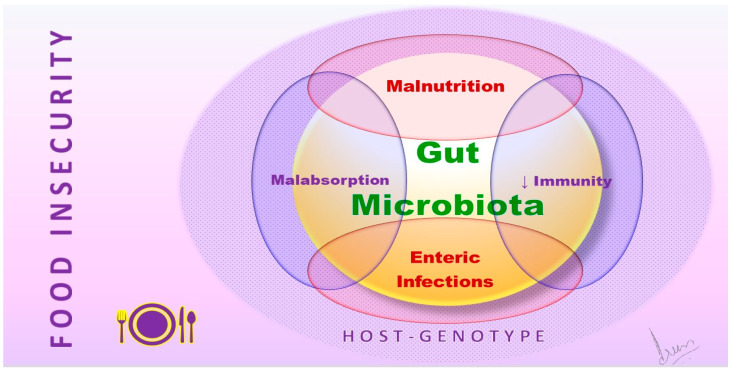
Complex multidirectional interactions of gut microbiota with major influencing factors of the unique host genotype and core modulation by the innate and adaptive host immunity, in the backdrop of food insecurity relevant to the host.

**Figure 2 nutrients-16-00005-f002:**
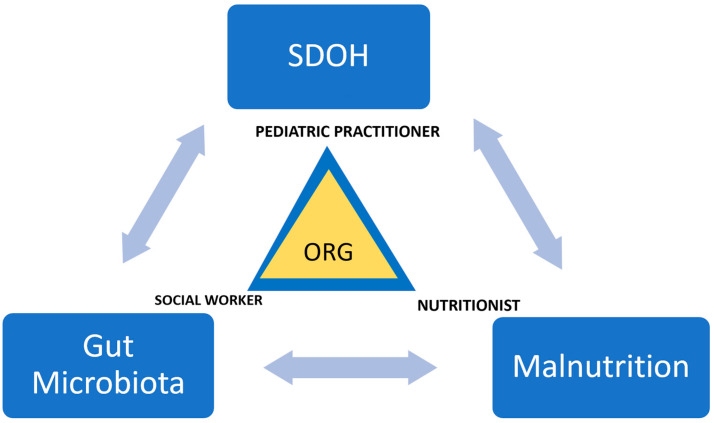
Interrelations of SDOH, Gut microbiota, and malnutrition. The role of pediatric practitioner, social worker, and nutritionist at the organizational level. ORG = Organization.

**Figure 3 nutrients-16-00005-f003:**
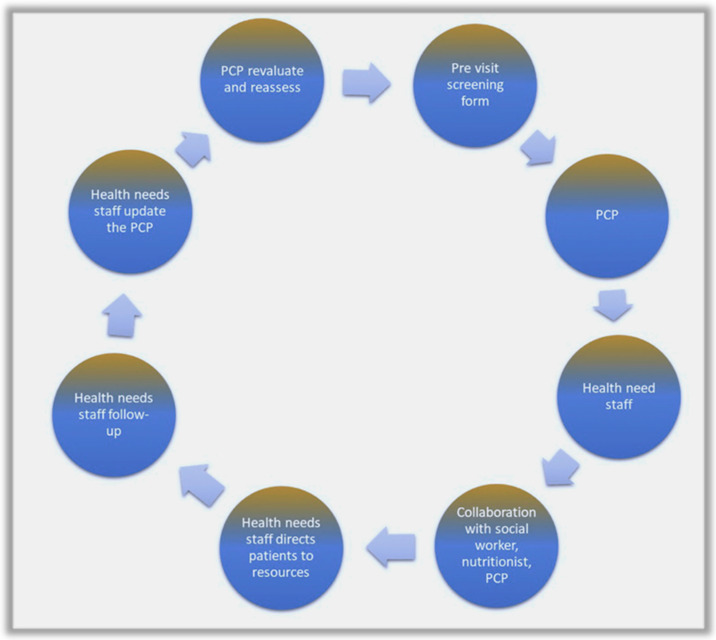
SSRR (Surveillance, Screening, Referral, and Reevaluation) model. PCP—Primary care Practitioner.

**Table 2 nutrients-16-00005-t002:** After Chung EK et al. and Network SIRaE. Social Needs Screening Tools Comparison Table (Pediatric Settings) [19,39].

Tools	Basic Description
**Child Trauma Questionnaire (CTQ)**	Child Maltreatment—emotional abuse, physical abuse, sexual abuse, emotional neglect, physical neglect
**History of Victimization Form**	Child maltreatment—sexual abuse, physical abuse, neglect, witness to family violence, and psychological abuse
**Kempe Family Stress Inventory**	Child Maltreatment—physical, sexual, and/or psychological abuse
**US Department of Agriculture Household Food Security Module**	Family Financial support—Assessing general household and child food security, categorized into high, marginal, low, and very low food security
**Two-question screen validated for clinical use**	Family Financial support—“Within the past 12 months we worried whether our food would run out before we got money to buy more.” “Within the past 12 months, the food we bought just didn’t last and we didn’t have money to get more.”
**HITS (Hurt, Insult, Threaten, Scream) tool**	Intimate partner violence, Conflict Tactics Scale (CTS) used as gold standard
**PVS (Partner Violence Screen)**	Intimate partner violence. CTS is used as a gold standard. A positive response to any item represents a positive screen
**WAST (Women Abuse Screening Tool), WAST-SF (Women Abuse Screening Tool—Short Form**	Intimate partner violence
**Patient Health Questionnaire-9 (PHQ-9)**	Maternal Depression and Family Mental Illness
**Edinburg Postnatal Depression Scale (EPDS)**	Maternal Depression and Family Mental Illness
**Safe Environment for Every Kid (SEEK)**	Household and Substance Abuse, Parent screening questionnaire
**Survey of Well-Being of Young Children (SWYC)**	Household Substance Abuse, Family Questionnaire
**HEADS (Home, Education & Employment, Activities, Drugs, Sexuality, Suicide/Depression)**	Household Substance Abuse
**CRAFT (Car, Relax, Alone, Forget, Friends, Trouble)**	Household Substance Abuse
**TOFHLA (Test of Functional Health Literacy in Adults)**	Parental Health Literacy
**REALM (Rapid Estimate of Adult Literacy in Medicine)** **REALM-R–shortened, revised version**	Patients are asked to read aloud to check for correct pronunciation.
**AHC—Tool Accountable Health Communities tool** **AHS HRSN Tools—Accountable Health Communities, Health-Related Screening Tools**	identify patient needs that can be addressed through community services in 4 domains (economic stability, social & and community context, neighborhood & and physical environment, and food)
**Health Leads**	Questionnaire assessing needs in 5 domains (economic stability, education, social & and community context, neighborhood & and physical environment, and food).
**MLP iHELLP**	Questionnaire assessing needs across 5 domains (economic stability, education, social & and community context, neighborhood & physical environment, and food).
**PRAPARE Protocol for Responding to & Assessing Patients’ Assets, Risks & Experiences**	It measures five domains of social determinants of health (SDOH): Housing status, Language, Employment, Transportation, Stress, Income.
**WellRx**	Questionnaire assessing needs in 4 domains (economic stability, education, neighborhood, physical environment, and food)
**Your Current Life Situation (YCLS)**	Questionnaire assessing needs in 6 domains (economic stability, education, social and community context, health and clinical care, neighborhood and physical environment, and food).
**We Care**	Questionnaire assessing needs in 4 domains (economic stability, education, neighborhood and physical environment, and food).

## Data Availability

Not applicable.

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
