# Peer review of "Role of Effective Policy and Screening in Managing Pediatric Nutritional Insecurity as the Most Important Social Determinant of Health Influencing Health Outcomes"

_nutrients, 2023, doi:10.3390/nu16010005_

Round 1

Reviewer 1 Report

Comments and Suggestions for Authors

Dear editors, it has been my pleasure to be invited to review this manuscript: Role of Effective Policy and Screening in Managing Pediatric Nutritional Insecurity as the Most Important Social Determinant of Health Influencing Health Outcomes’. Here are my comments:

Comments 1:

Line 7: SLU College For Public Health and Social Justice, Saint Louis University, Saint Louis, MO, 63104 7

This line lacks the country where the institution is located, please add it.

Comments 2:

Line 33: According to Healthy People Framework 2030, conceived by the US National Secretary’s Advisory Committee on National Health Objectives for the year 2030, SDOH can be defined as “The conditions in the environments where people are born, live, learn, work, play, worship, and age that affects a wide range of health, functioning, and quality-of-life outcomes and risks.

What does SDOH stand for? In the main body of the manuscript, please write out the full spelling of the abbreviation for the first time.

Comments 3:

Line 32-36: According to Healthy People Framework 2030, conceived by the US National Secretary’s Advisory Committee on National Health Objectives for the year 2030, SDOH can be defined as “The conditions in the environments where people are born, live, learn, work, play, worship, and age that affects a wide range of health, functioning, and quality-of-life outcomes and risks.

Please add specific references to support your statement.

Comments 4:

Line 36-38: The SDOH has been categorized into five domains: Economic Stability, Education Access and Quality, Health Care Access and Quality, Neighborhood and Built Environment, and Social and Community Context [1)].

What does the "[1)]" that appears at the end of this sentence mean? I would suggest that this article be proofread for formatting once by a professional editor.

Comments 5:

Line 49-52: As indicated by an official evaluation by the Office of Assistant Secretary for Planning and Evaluation of the US Government, merely 20% of health disparities at the county level stem from clinical care, whereas SDOH contributes to around 50% of health results.

Please add specific references to support your statement.

Comments 6:

Line 54-56: Another report by the Economic Research Service of the US Department of Agriculture noted that addressing food insecurity is crucial in improving population health, as it is a major social determinant of well-being.

Another report? You should provide a reference to support where this statement comes from.

Comments 7:

Line 56-58: Therefore, hospitals must consider behavioral, socio-economic, and environmental factors when implementing a comprehensive health strategy [4].

Is this a problem that hospitals can solve? Does it seem to require the involvement of other stakeholders?

Comments 8:

Line 61-62: After Byhoff E et al [5]

Where did that quote come from? What is the purpose of it? Please proofread the manuscript again.

Comments 9:

Line 151-155: Social Determinants of Health (SDOH) wield an impact on the well-being of individuals spanning various age groups; however, their significance is notably pronounced for children and young individuals. This is because the foundational elements for lifelong health and well-being, encompassing physical, social, and emotional capacities, are established during the early stages of life. [16].

Why does the full spelling and abbreviation of SDOH appear here again?

Comments on the Quality of English Language

Revise

Author Response

Dear Reviewer,

Thank you very much for thoroughly critiquing our manuscript. Your recommendations have been extremely helpful. Please find below a point-by-point response to the reviewer comments. Each critique is followed by a ‘Response’ in red indicating our action.

The manuscript has been proofread again by first and second author and all formatting and citations are confirmed. English language revised by self and using assisting software like MS Word and Grammarly for grammar, syntax, and style.

If any section or subsection needs further revision, please let us know and we shall promptly comply.

Sincerely,

Hema Verma

Arun Verma

Comments 1:

Line 7: SLU College For Public Health and Social Justice, Saint Louis University, Saint Louis, MO, 63104 7

This line lacks the country where the institution is located, please add it.

Response: Country name added.

Comments 2:

Line 33: According to Healthy People Framework 2030, conceived by the US National Secretary’s Advisory Committee on National Health Objectives for the year 2030, SDOH can be defined as “The conditions in the environments where people are born, live, learn, work, play, worship, and age that affects a wide range of health, functioning, and quality-of-life outcomes and risks.

What does SDOH stand for? In the main body of the manuscript, please write out the full spelling of the abbreviation for the first time.

Response: Full form of SDOH added in the beginning

Comments 3:

Line 32-36: According to Healthy People Framework 2030, conceived by the US National Secretary’s Advisory Committee on National Health Objectives for the year 2030, SDOH can be defined as “The conditions in the environments where people are born, live, learn, work, play, worship, and age that affects a wide range of health, functioning, and quality-of-life outcomes and risks.

Please add specific references to support your statement.

Response: Reference added

Comments 4:

Line 36-38: The SDOH has been categorized into five domains: Economic Stability, Education Access and Quality, Health Care Access and Quality, Neighbourhood and Built Environment, and Social and Community Context [1)].

What does the "[1)]" that appears at the end of this sentence mean? I would suggest that this article be proofread for formatting once by a professional editor.

Response: Formatting corrected and reviewed in the entire paper.

Comments 5:

Line 49-52: As indicated by an official evaluation by the Office of Assistant Secretary for Planning and Evaluation of the US Government, merely 20% of health disparities at the county level stem from clinical care, whereas SDOH contributes to around 50% of health results.

Please add specific references to support your statement.

Response: Reference added and hyperlink introduced in bibliography

Comments 6:

Line 54-56: Another report by the Economic Research Service of the US Department of Agriculture noted that addressing food insecurity is crucial in improving population health, as it is a major social determinant of well-being.

Another report? You should provide a reference to support where this statement comes from.

Response: Reference added

Comments 7:

Line 56-58: Therefore, hospitals must consider behavioral, socio-economic, and environmental factors when implementing a comprehensive health strategy [4].

Is this a problem that hospitals can solve? Does it seem to require the involvement of other stakeholders?

Response: Hospitals are one of the many important stakeholders in the screening of SDOH with questionnaires and standard screening forms already available on government websites free of cost. The deficiency lies in the fact that these measures are not followed, the involvement of stakeholders at the state or federal level in developing policies can have a significant impact. Details are available in subsequent sections.

Comments 8:

Line 61-62: After Byhoff E et al [5]

Where did that quote come from? What is the purpose of it? Please proofread the manuscript again.

Response:After Byhoff et alis not a comment. It is the label of the accompanying figure. Ref. No [5] is listed with it as the source.

Comments 9:

Line 151-155: Social Determinants of Health (SDOH) wield an impact on the well-being of individuals spanning various age groups; however, their significance is notably pronounced for children and young individuals. This is because the foundational elements for lifelong health and well-being, encompassing physical, social, and emotional capacities, are established during the early stages of life. [16].

Why does the full spelling and abbreviation of SDOH appear here again?

Response: Abbreviations used, and full form deleted.

Reviewer 2 Report

Comments and Suggestions for Authors

Review of "Role of Effective Policy and Screening in Managing Pediatric Nutritional Insecurity as the Most Important Social Determinant of Health Influencing Health Outcomes" by Hemy Verma, Arun Verma, Jeffrey Bettag, Sree Kolli, Kento Kurashima, Chandrashekar Manithoda and Ajay Jaina.

Strengths:

1.The paper presents an in-depth analysis of social determinants of health (SDOH) with a focus on children, demonstrating a solid theoretical understanding.

2.The authors aptly identify pediatric risks associated with unfavorable social conditions, such as malnutrition, gut microbiome disorders, and acute and chronic diseases.

3 The paper presents a coherent SSRR plan that covers the full spectrum of social determinants of health, and particularly focuses on the problem of child malnutrition. Proposals for nutrition and education recommendation programs are well thought out.

Weaknesses:

1.The work is based mainly on theoretical analysis, but lacks concrete empirical data, which weakens the strength of the conclusions and proposals.

2. The authors note shortcomings in existing programs, but do not provide a deeper analysis, making it difficult to understand why these programs are ineffective.

3. The paper focuses on the SSRR plan, but does not analyze potential adverse effects or contraindications, which may be important for public health practitioners.

4. Adaptation of the literature and editing according to MDPI guidelines.

Recommendations to authors:

1. To strengthen the argument, it is recommended to add empirical data, such as survey results or health statistics, to support the theses presented.

2. The authors should conduct a more detailed analysis of current nutrition programs to better understand why they are ineffective, so that changes can be proposed more effectively.

3. it is important to include possible contraindications to the proposed SSRR plan to avoid potential negative effects in practice.

4. Detailn strengths and weaknesses after discussion. 

In conclusion, the paper presents an important issue of social determinants of health for children, but could be strengthened with concrete empirical data and more detailed analysis of existing nutrition programs. Implementation of the suggested recommendations could improve the health of children and their communities.

Author Response

Dear Reviewer,

Thank you very much for thoroughly critiquing our manuscript. Your recommendations have been extremely helpful.

Please find below a point-by-point response to the reviewer comments. Each critique is followed by a ‘Response’ in red indicating our action.

The manuscript has been proofread again by first and second author and all formatting and citations are confirmed. English language revised by self and using assisting software like MS Word and Grammarly for grammar, syntax, and style.

If any section or subsection needs further revision, please let us know and we shall promptly comply.

Sincerely,

Hema Verma

Arun Verma

Strengths:

1.The paper presents an in-depth analysis of social determinants of health (SDOH) with a focus on children, demonstrating a solid theoretical understanding.

Response: Thank you very much.

2.The authors aptly identify pediatric risks associated with unfavorable social conditions, such as malnutrition, gut microbiome disorders, and acute and chronic diseases.

Response: Thank you very much.

  1. The paper presents a coherent SSRR plan that covers the full spectrum of social determinants of health, and particularly focuses on the problem of child malnutrition. Proposals for nutrition and education recommendation programs are well thought out.

Response: Thank you very much.

Weaknesses:

1.The work is based mainly on theoretical analysis, but lacks concrete empirical data, which weakens the strength of the conclusions and proposals.

Response:

We have used publicly available national and global data as per publications referenced in the body of the paper. The data was not tabulated but used in line with text to maintain continuity of discussion. Examples are ref. no. 3, 7, 10, 11, 12, 14 and so on.

  1. The authors note shortcomings in existing programs, but do not provide a deeper analysis, making it difficult to understand why these programs are ineffective.

Response: SDOH are only recently emerging as important parameters of health outcomes, especially in children. There is often a lack of political willpower or short-sightedness of policy framing and implementation and then, there is an issue of appropriate utilization of funds allocated for health and nutritional policy reform process. Correct policy framework and policy implementation also requires multi-institutional coordination over a period of decades [Encompassing ever changing political scenario] and therefore the process must be institutionalized via law enactment and proactive participation of the healthcare framework, educational institutes and other stakeholders.

  1. The paper focuses on the SSRR plan, but does not analyze potential adverse effects or contraindications, which may be important for public health practitioners.

Response: SSRR plan is a novel concept and meant to integrate nutrition-based Surveillance, Screening, Referral, and Reevaluation (SSRR) into an institutional framework and then, to utilize policy, laws, nutritional programs, and educational & healthcare stakeholders to promote positive health outcomes. We will be able to evaluate potential roadblocks and improve on the concept in our next phase of study.

  1. Adaptation of the literature and editing according to MDPI guidelines.

Response: We have reviewed the paper again for MDPI guidelines.

Thank you for your additional recommendations. We will take them into account.

Response: After this review paper, we plan a second phase of our research where we will conduct clinical research to generate our own data in local geographical domain, which will allow us to tabulate existing SDOH disparities, evaluate their relationships to health outcomes, and then utilize specific interventions to study their impact on amelioration of those health outcome differences.

Round 2

Reviewer 1 Report

Comments and Suggestions for Authors

Addressed the comments